# Effects of Feeding Garlic Powder on Growth Performance, Rumen Fermentation, and the Health Status of Lambs Infected by Gastrointestinal Nematodes

**DOI:** 10.3390/ani9030102

**Published:** 2019-03-20

**Authors:** Rongzhen Zhong, Hai Xiang, Long Cheng, Chengzhen Zhao, Fei Wang, Xueli Zhao, Yi Fang

**Affiliations:** 1Jilin Provincial Key Laboratory of Grassland Farming, Northeast Institute of Geography and Agroecology, Chinese Academy of Sciences, Changchun 130102, China; zhongrongzhen@iga.ac.cn (R.Z.); xianghai18@mails.ucas.ac.cn (H.X.); zhaochengzhen@iga.ac.cn (C.Z.); wangfei@iga.ac.cn (F.W.); zhaoxueli6367@163.com (X.Z.); 2CAS Key Laboratory for Agro-Ecological Processes in Subtropical Region, Institute of Subtropical Agriculture, the Chinese Academy of Sciences, Changsha 410125, China; 3University of Chinese Academy of Sciences, Beijing 100049, China; 4Faculty of Veterinary and Agricultural Sciences, Dookie Campus, the University of Melbourne, Victoria 3647, Australia; long.cheng@unimelb.edu.au

**Keywords:** *Allium sativum*, sheep, feed efficiency, parasite, bioactive compounds

## Abstract

**Simple Summary:**

Non-chemical strategies to control gastrointestinal nematodes (GINs) infection are urgently needed to support the sustainable development of the livestock industry. The potential anti-parasitic properties in garlic powder to control sheep GINs was investigated in this study. The key finding of this study was that feeding garlic powder increased growth performance of lambs infected with GINs by decreasing fecal egg counts and improving feed digestion, rumen fermentation, and the health status of lambs.

**Abstract:**

For the study, forty lambs were weighed and assigned into two treatments to determine the effects of feeding garlic powder on growth performance, rumen fermentation, and the health status of lambs infected with gastrointestinal nematodes (GINs). The lambs were fed with a basal diet without or with 50 g/kg garlic powder for 84 d. Data were analyzed by a general linear or mixed model of SAS software and differences were considered statistically significant if *p* ≤ 0.05. Results showed that garlic powder supplementation increased the lambs’ average daily gain (*p* = 0.025), digestibility of dry matter (*p* = 0.019), and crude protein (*p* = 0.007). No significant changes were observed on the dry matter intake, feed conversion ratio, as well as the apparent digestibility of lipid and fiber. An interactive effect between treatment and feeding day was observed on ruminal pH (*p* < 0.001) and ammonia nitrogen concentration (*p* < 0.001). Ruminal pH (*p* < 0.001) and ammonia nitrogen concentration (*p* < 0.001) decreased with garlic powder supplementation, while ammonia nitrogen concentration increased (*p* = 0.001) with the extension of the feeding period. Garlic powder supplementation increased the total volatile fatty acid concentration (*p* < 0.001) in the rumen fluid, the molar proportion of acetic acid (*p* = 0.002), propionic acid (*p* < 0.001), and isovaleric acid (*p* = 0.049), but it decreased the ratio of acetic acid to propionic acid (*p* = 0.015). The lambs’ fecal egg count decreased (*p* < 0.001), but the packed cell volume and body condition scores of lambs increased (*p* < 0.001) with garlic powder supplementation. In conclusion, feeding garlic powder increased growth performance, feed digestion, rumen fermentation, and the health status of lambs infected with GINs.

## 1. Introduction

Gastrointestinal nematodes (GINs) infection is a major constraint to sheep and goat production [1]. *Haemonchus contortus* is a major type of blood-sucking parasite in sheep and goats, which can lead to anaemia and production loss [2]. Over the past 20 years, over-dependency and misuse of chemical anthelmintic drugs to control GINs infections has led to anthelmintic drug resistance in livestock products. Thus, alternative GINs control strategies are needed [3]. 

Garlic (*Allium sativum*) has antimicrobial, antioxidant, as well as antihypertensive properties, and has been used as a feed flavor in the livestock industry [4,5]. Apart from these properties, some recent studies revealed that garlic extracts have anti-parasitic properties against *Aspiculuris tetrapte* infection in mice [6], *Neobenedenia sp.* infection in fish [7], GIN infection in goats [8], and *Trypanosoma b. brucei* and *Leishmania tarentolae* infection in humans [9]. It has been widely reported that several bioactive sulfur-containing compounds in garlic, such as alliin, diallylsulfides, and allicin exert anti-parasitic activity due to their special chemical structures, which can interact with sulfhydryl groups of proteins to block the physiological metabolism of parasites [10,11]. However, bioactive compounds extraction protocols are generally complicated and costly, which limit the wide use of dietary garlic oil and pure allicin as a supplement in the livestock industry to control GINs. Therefore, supplementing garlic powder, which contains all bioactive components and is prepared through simple steps, may be an alternative and practical strategy to deliver bioactive compounds to livestock. Furthermore, previous work conducted to examine the usefulness of garlic/garlic extract as a supplement in livestock production focused on growth performance [12], health [13], or in vitro rumen function [14]. For all we know, there is limited research investigating the supplementation of garlic powder or extract effect on sheep-infected GINs, which includes growth, digestion, rumen metabolism, and health status within one study.

The objective of this study investigates the effects of dietary garlic powder supplementation on growth performance, rumen fermentation, and health status, including body condition score (BCS) and packed cell volume (PCV), of growing lambs infected with GINs.

## 2. Materials and Methods

### 2.1. Garlic Powder Preparation

Fresh garlic was purchased from a market and garlic powder was prepared by drying garlic under the sun at ~24.5 °C for 3 d, and then ground to pass through a 4 mm sieve. Garlic powder contained: Dry matter (DM) (465.4 g/kg of the feed), crude protein (CP) (18.15% of DM), lipid (0.53% of DM), neutral detergent fiber (NDF) (7.52% of DM), acid detergent fiber (ADF) (5.37% of DM), and ash (5.37% of DM).

### 2.2. Animals and Experiment Design

The ethics approval was completed by the Animal Care Committee of the Institute of Geography and Agroecology, Chinese Academy of Sciences, Jilin, China (Protocol No. 2017003).

Forty male Ujumqin lambs with an average age of (mean ± SD) 160 ± 10.5 d old and body weight (BW) of 26 ± 1.2 kg were randomly assigned to two groups. This resulted in 20 lambs per group. The study consisted of a 14 d diet adaptation period and an 84 d animal measurement period. Lambs in control treatment (Table 1) were fed a basal diet without garlic powder supplementation, while lambs in the experimental group were fed a basal diet supplemented with 50 g garlic powder/kg of the feed (DM basis). Both diets were formulated according to the nutrient requirements of sheep [15]. All ingredients except for *Aneurolepidium* Chinese hay and maize straw were totally mixed and pelleted. The pelleting machine (model: GSL) was purchased from Changzhou Ou Drying Equipment Co., Ltd. (Changzhou, China). During the pelleting process, steam was added and a 75 °C exit temperature was achieved and the machine yielded an average of 1.0 tons/h with a pellet diameter of 2.5 mm. Adequate feed was produced in one batch for the entire experiment.

All lambs were dewormed using a combination anthelmintic of abamectin (0.2 mg/kg BW), levamisole (7.5 mg/kg BW), and albendazole (5 mg/kg BW) (Pyrimide^®^, Novaritis Animal Health Co., Ltd., Shanghai, China) to eliminate existing GINs 14 d prior to the start of the feed adaptation period. After deworming, all lambs were housed individually and spot fecal samples were collected weekly to check for fecal egg counts (FEC) using the modified McMaster’s technique [16] with results expressed as eggs per gram (EPG) of feces to ensure no presence of GINs at the start of the measurement period. On d 0 of the measurement period prior to feeding, all lambs were experimentally drenched with approximately 10,000 GIN third-stage larvae, which were hatched by the egg hatching procedure [17] in one batch. The GIN species that implanted in the rumen were 0.78 ± 0.020 *H. contortus*, 0.16 ± 0.030 *Trichostrongylus colubriformis*, 0.03 ± 0.005 *Ostertagia circumcincta*, and some other species of nematodes identified by light microscopy (Bausch & Lomb, Scientific Optical Products Div., Rochester, NY, USA).

During the entire measurement period, lambs were reared in the same pen and confined individually in metabolic cages (140 cm × 100 cm × 124 cm) with ad libitum access to water and feed provided every day at 07:00 and 18:00.

### 2.3. Sampling and Analysis Procedure

Feed intake was calculated from feed offered and refused daily. The DM of feed samples was determined by drying samples to a constant weight at 105 °C. Prior to the morning feeding of measurement d 0, 28, 56, and 84, all lambs were weighed and the average daily gain (ADG) per lamb was calculated as the regression of BW measured over time. The feed conversion ratio (FCR) was derived by dividing the DM intake (DMI) by the ADG. 

Before the morning feeding of d 0, 28, 56, and 84 of the feeding period, the BCS of each lamb (1 to 5, with 1 and 5 indicating very thin and fat, respectively) was determined according to the method described by [18], rectal fecal sample of each lamb was collected to determine FEC using a McMaster technique [19], and subsequently a 7 mL blood sample for each lamb was collected from the jugular vein by venipuncture into a heparin sodium-containing tube (Becton-Dickinson, Vacutainer Systems, Rutherford, NJ, USA) to measure PCV using a Unico micro-haematocrit centrifuge (182-E, Dayton, NJ, USA) and a micro-capillary reader (Damon/IEC Division). 

Rumen fluids were sampled per lamb using oral lavage per method of [20] at 2 h after the morning feeding on measurement d 0, 28, 56, and 84. pH of rumen fluid was measured using a pH meter (Testo 205, Testo AG, Lenzkirch, Germany). Then, rumen fluid samples were strained through 4 layers of cheesecloth and centrifuged at 10,000× *g* for 15 min at 4 °C. The samples, 15 mL, were transferred into tubes containing 0.3 mL of 500 mg/g sulfuric acid, mixed and stored at −20 °C for ammonia nitrogen (N) analysis by the phenol-hypochlorite method [21]. The remaining 10 mL strained samples were mixed with 1 mL of 250 mg/g metaphosphoric acid and stored at −20 °C for total and individual volatile fatty acid (VFA) determination by gas chromatography [22].

The digestibility trial consisted of a 5 d metabolic cage adaption period, and a 7 d period for total feces collection. The total collection of feces allowed the calculation of apparent digestibility (DM and nutrients) between d 70 and 76 of the measurement period. Daily collected fresh feces were subsampled (100 g/kg) and stored at −20 °C. Samples per lamb for 7 d were thawed and pooled (500 g per lamb), and dried at 105 °C to determine DM content. All feeds and the remaining feces samples were dried at 65 °C for 48 h, ground to pass through a 1 mm screen (Wiley Mill, Arthur H. Thomas, Philadelphia, PA, USA). The total ether extract [23] and N [24] of diets and feces were measured. The content of NDF and ADF were determined by the method of [25] using heat-stable alpha-amylase and sodium sulfite for NDF determination.

### 2.4. Statistical Analysis

Data on digestibility of feed DM and nutrients, as well as the lambs’ growth performance were analyzed using a general linear model [26]. Data of blood, BCS, and FEC were analyzed by the MIXED model procedure of SAS 8.1 (SAS Institute Inc., Cary, NC) [27]. The model consisted of dietary treatment, sampling day, dietary treatment × sampling day interactions as fixed effects. The random effect was animal. Measurements obtained per lamb at different sampling days were treated as repeated measures. Differences were considered statistically significant if *p* ≤ 0.05. 

## 3. Results

### 3.1. Effect of Garlic Powder on DMI, ADG, FCR, and Digestion

Dietary garlic powder supplementation increased ADG (*p* = 0.025) but did not affect DMI and FCR of lambs infected with GINs (*p* > 0.05) (Table 2). The apparent digestibility of feed DM (*p* = 0.019) and CP (*p* = 0.007) increased by garlic powder supplementation. However, the apparent digestibility of lipid, NDF, and ADF did not change by garlic powder supplementation. 

### 3.2. Effect of Garlic Powder on Rumen Fermentation

Table 3 showed that the interaction effects between treatment and feeding time were found on ruminal pH (*p* < 0.001) and ammonia N concentration (*p* < 0.001). Ruminal pH of lambs fed with garlic powder was lower (*p* < 0.001) than that of CON lambs. Dietary garlic powder supplementation increased the total VFA (*p* < 0.001), acetic acid (*p* = 0.002), propionic acid (*p* < 0.001), and isovaleric acid (*p* = 0.049) concentration, but decreased the ratio of acetic acid and propionic acid (*p* = 0.015). Different feeding times had no effect on VFA pattern of lambs (Table 3). 

### 3.3. Effect of Garlic Powder on FEC, PCV, and BCS

Dietary garlic powder supplementation decreased FEC (*p* < 0.05) of lambs and the degree of effect enhanced with the prolongation of the measurement period from d 28 to 84 (Figure 1a). The PCV values of lambs infected with GINs increased (*p* < 0.001) by dietary garlic powder supplementation (Figure 1b). The BCS values of lambs on d 28 of the measurement period increased (*p* = 0.005) and this effect was enhanced (*p* < 0.001) from d 28 to 84 of the measurement period (Figure 1c).

## 4. Discussion

The GINs infection in small ruminants have been largely found to decrease ADG by 23%–63% [28] of host animals. Furthermore, previous work showed GINs reduced DMI [29], feed digestion [30], and meat quality [31], and increased FCR [32] of host animals. Though having the above negative effects on the host animals, the main economic losses caused by GINs infection are not from the mortality rate, although up to 25% pre-weaning mortality was reported, but from ADG loss [3,28]. Thus, any practical strategies to decrease parasitic load and increase ADG will be beneficial to profitability in livestock production. The effects of feeding garlic or garlic extracts on the ADG and DMI livestock were not consistent in the literature. For example, Chaves et al. (2008) reported that dietary garlic oil supplementation had no effects on DMI and ADG of lambs without parasitic infection [33], whereas Hasan et al. (2015) found that the ADG of grazing goats infected with internal parasites increased 10.3% when goats were fed with a water solution of garlic [8]. The present results showed that the ADG of lambs infected with GINs significantly increased by 9.5% when the lambs were fed with garlic powder. Although DMI increased by 5.4% and FCR decreased by 4.3%, the changes did not reach to a significant level. Future work is needed to understand the reasons behind the observed discrepancy in ADG between studies with small ruminants. 

Rumen fermentation of ruminants would change after infection by GINs due to the complex interplay between GINs and host commensal flora [34]. Sahli et al. (2018) did not find any changes in in vitro rumen fermentation by including garlic powder [14], while Busquet et al. (2005a) revealed that garlic oil feeding decreased the molar proportion of acetate and increased the proportions of propionate and butyrate in an in vitro fermented system [35]. Our in vivo study found that dietary garlic powder supplementation increased total VFA content and the proportion of acetate, propionate, and isovalerate, but reduced the ratio of acetate to propionate. This is partly consistent with the findings of [36,37], in which they reported that garlic oil increased the proportions of propionate and butyrate but reduced the proportion of acetate. Nevertheless, a reduction in the ratio of acetate to propionate in rumen indicates an improvement in animal net energy status [38] and hence it may contribute to a higher ADG in our study. 

The FEC is used routinely to indicate GINs infection in ruminants, which correlates well with worm burden and represents the best phenotypic marker for GINs infection [39,40]. Consistent with previous works, which reported that garlic extract possessed inhibition properties in parasite infection [41,42], our present results showed that garlic powder feeding decreased FEC of lambs and the impact was strengthened with the prolongation of feeding time. The decreased FEC might be due to decreased fecundity of GINs by garlic powder supplementation, which has been suggested by [43]. In addition, the reduced FEC might also be attributed to those bioactive compounds that might kill adult worms in both direct and indirect ways based on the conclusions of [44]. The direct action mode of anthelmintic garlic powder might be attributed that thiosulfinate sulfur damaged the physiological processes of nematodes and then reduced fecundity and worm burden [45], while indirect action mode might be attributed to that allicin in garlic powder acted on various immune-correlated processes in animals and even in humans [46]. 

A negative correlation was identified between *H. contortus* burden and PCV in blood [47]. The *H. contortus* is haematophagus parasite with a high egg-lying capacity, which may cause severe blood loss in the host animals, resulting in anemia, anorexia, depression, and even death [2]. This is consistent with the findings of [13], in which they reported that feeding garlic extract dramatically increased PCV of goat kids. The current results indicated that the PCV of lambs increased due to garlic powder supplementation. Increased PCV could be mainly due to reduced FEC or due to improved nutrient supply by garlic powder supplementation (i.e., the supplemented group diet had 4% and 8% higher CP and ME than the non-supplemented group diet), allowing the animals to cope with a high GIN burden and to replace lost blood cells [48,49]. 

The BCS is a method for the subjective assessment of an animal’s health status based on the estimation of their body fatness and muscle. It is considered as a valid welfare indicator in sheep [50] and goats [51]. Healthy sheep and goats should have a BCS of 2.5 to 4.0 [52]. The present results indicated that the BCS decreased sharply by GINs infection without garlic powder supplementation, whereas it ranged from 4.0 to 4.5 in garlic powder supplemented sheep, which reflects an improved health status of lambs supplemented with garlic powder vs. non-supplemented sheep. The improved BCS might be partly due to decreased FEC of lambs, as a negative correlation between BCS and FEC was widely found [53,54]. In addition, the increased dietary CP intake by garlic powder feeding also may have contributed to the nutritional and health enhancement of lambs, because more muscle may be deposited when the GINs burden decreased [55]. 

## 5. Conclusions

The results of this study indicate that dietary garlic powder supplementation increased the growth performance of lambs infected with GINs by decreasing FEC and improving feed digestion, rumen fermentation, and the health status of lambs. Further studies are needed to test different supplementation levels of garlic powder and its impact on sheep productivity.

## Figures and Tables

**Figure 1 animals-09-00102-f001:**
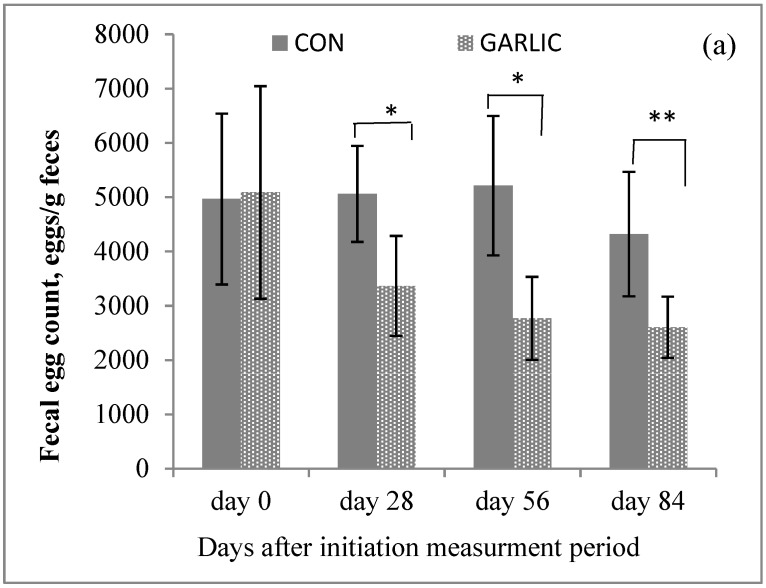
Fecal egg counts: (**a**) Packed cell volume, (**b**) body condition scores, and (**c**) (mean ± standard error) of lambs infected with gastrointestinal nematodes and fed with or without garlic powder. CON: Control treatment; GARLIC: Garlic powder supplemented treatment. *: *p* < 0.05; **: *p* < 0.01.

**Table 1 animals-09-00102-t001:** Ingredients and chemical composition of the experiment’s diets.

Item	Treatment	
	CON	GARLIC
Ingredients, %		
*Aneurolepidium* Chinese hay	28.0	28.0
Maize straw	10.0	5.0
Garlic powder	0	5.0
Ground corn grain	45.0	45.0
Soybean bean	14.0	14.0
Dicalcium phosphate	0.5	0.5
Sodium chloride	1.0	1.0
Limestone	0.5	0.5
Minerals and Vitamins salt *	1.0	1.0
Chemical composition (n = 6)		
Dry matter (DM), % of feed,	89.2	90.1
Crude protein, % of DM	14.5	15.1
Neutral detergent fiber, % of DM	40.2	38.4
Acid detergent fiber, % of DM	20.8	18.6
Lipid, % of DM	6.8	7.3
Metabolizable energy, ^†^ Mcal/kg DM	10.6	11.4

* The mineral salt and vitamins were purchased from Continental Grain Crop (Beijing, China) and contained (per kg): 16.8 g Ca, 8.0 g P, 12.0 g Na, 1.5 g Mg, 1.5 g K, 1.5 g S, 1.25 g Fe, 1.25 g Mn, 1.25 g Z, 240 mg Co, 1750 mg Cu, 450 mg I, 50 mg Se, 300,000 IU/Ib vitamin A, 50,000 IU/Ib vitamin D3, and 400 IU/Ib vitamin E. ^†^ Metabolizable energy was estimated from nutrient requirements of small ruminants [16].

**Table 2 animals-09-00102-t002:** Effects of garlic powder supplementation on body weight gain, dry matter intake, and feed digestion of lambs infected with gastrointestinal nematodes.

Index	Treatment *		SEM	*p* Value
	CON	GARLIC		
DMI, g/d	923	973	22.7	0.129
CPI, g/d	90.1	109.2	3.11	<0.001
ADG, g/d	221	242	6.3	0.025
FCR (DMI/ADG)	4.22	4.04	0.151	0.396
Apparent digestibility, %				
DM	60.29	64.21	1.088	0.019
CP	68.27	74.38	1.687	0.007
Ether extract	69.97	72.28	1.423	0.263
NDF	43.41	44.87	0.675	0.140
ADF	30.80	32.65	1.123	0.255

* CON: Control treatment; GARLIC: Garlic powder supplemented treatment; DMI: Dry matter intake; CPI: Crude protein intake; ADG: Average daily gain; FCR: Feed conversion ratio; CP: Crude protein; NDF: Neutral detergent fiber; ADF: Acid detergent fiber.

**Table 3 animals-09-00102-t003:** Effects of dietary garlic powder supplementation on rumen fermentation parameters of lambs infected with gastrointestinal nematodes.

Index ^2^	Treatment *		SEM	Feeding Day				SEM	*p* Value		
	CON	GARLIC		0	28	56	84		Treat (T)	Day (D)	T × D
Ruminal pH	6.55 ^a^	6.18 ^b^	0.026	6.40	6.36	6.36	6.34	0.037	<0.001	0.710	<0.001
Ammonia N, mmol/L	15.35 ^a^	12.60 ^b^	0.256	12.97 ^b^	14.55 ^a^	14.85 ^a^	13.54 ^ab^	0.362	<0.001	0.001	<0.001
Total VFA, mmol/L	86.38 ^b^	90.79 ^a^	0.661	88.69	88.15	88.91	88.62	0.934	<0.001	0.950	0.196
VFA, mol/100 mol											
Acetic acid (A)	62.00 ^b^	64.21 ^a^	0.482	62.89	62.70	63.92	62.90	0.682	0.002	0.587	0.642
Propionic acid (P)	13.52 ^b^	15.52 ^a^	0.379	14.87	14.45	14.02	14.75	0.536	<0.001	0.679	0.316
A:P	4.78 ^a^	4.27 ^b^	0.146	4.45	4.44	4.84	4.38	0.206	0.015	0.384	0.327
Butyric acid	6.12	6.38	0.165	6.24	6.20	6.22	6.34	0.233	0.285	0.981	0.890
Isobutyric acid	1.64 ^a^	1.45 ^b^	0.042	1.53	1.60	1.57	1.48	0.060	0.003	0.526	0.789
Valeric acid	1.39	1.41	0.037	1.37	1.48	1.39	1.37	0.053	0.749	0.428	0.642
Isovaleric acid	1.71 ^b^	1.83 ^a^	0.044	1.78	1.72	1.79	1.78	0.062	0.049	0.828	0.133

Means values with different letters (^a,b^) in the same row within treatment or feeding time differ (*p* < 0.05). * CON: Control treatment; GARLIC: Garlic powder supplemented treatment; VFA: Volatile fatty acid.

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
