# Peer review of "Effects of Feeding Garlic Powder on Growth Performance, Rumen Fermentation, and the Health Status of Lambs Infected by Gastrointestinal Nematodes"

_animals, 2019, doi:10.3390/ani9030102_

Round 1

Reviewer 1 Report

All my comments were addressed carefully. Thanks

Reviewer 2 Report

Figure 1, panel a Fecal egg count is duplicated. Please remove the dublicate and update the numbering.

This manuscript is a resubmission of an earlier submission. The following is a list of the peer review reports and author responses from that submission.

Round 1

Reviewer 1 Report

1- The summary is too wordy. The English needs some revision.

2- In the abstract you need to mention how much lambs weigh?

3- In the abstract you need to mention for how long lambs were fed the diet?

4- In the abstract you need to mention the statistical analysis

5- In the abstract you need to mention what was the significant level

6- in the introduction section, mention how much production loss in caused by GINs? Do you have numbers that show the percentage of loss?

7-How many lambs were in each of control and experimental groups?

8- In statistical analysis you need to mention what the SAS version was?

9- Revise table 3 format. Some numbers are pushed to the next row.

10- Revise figure 1. Some of stars were moved to the wrong place.

11- In the discussion section mention how much decrease in ADG, DMI and feed digestion is caused by GINs/ Give some numbers.

12- In the discussion section mention how much increase in FCR is caused by GINs/ Give some numbers.

13- In the discussion section mention what is the mortality rate of GINs?

line 185 Revise figure 1. Some of stars were moved to the wrong place. 

line189 How much decrease in ADG, DMI and feed digestion? Give some numbers.
line 190 increase FCR: How much decrease in ADG, DMI and feed digestion? Give some numbers.

line 190 what is the mortality rate?

Author Response

Animals

March 5, 2019

Dear reviewer:

Thanks for your detailed comments and review time which was put into this manuscript. Our responses to each comment are as follows:

1-     The summary is too wordy. The English needs some revision.

Responses: The summary was revised and the English was improved.

2-     In the abstract you need to mention how much lambs weigh?

Responses: “weighed and” was added after “Forty lambs were”.

3-     In the abstract you need to mention for how long lambs were fed the diet?

Responses: “for 84 days” was added.

4-     In the abstract you need to mention the statistical analysis

Responses: revised as suggested.

5-     In the abstract you need to mention what was the significant level

Responses: The significant level was added.

6-     in the introduction section, mention how much production loss in caused by GINs? Do you have numbers that show the percentage of loss?

Responses: The numbers of weight lose of sheep and goats by GIN infection was added according to reference and the previous reference was removed.

7-     How many lambs were in each of control and experimental groups?

Responses: “(20 lambs in each group)” was added in line 83.

8-     In statistical analysis you need to mention what the SAS version was?

Responses: The version “8.1” was added.

9-     Revise table 3 format. Some numbers are pushed to the next row.

Responses: Revised as suggested.

10-  Revise figure 1. Some of stars were moved to the wrong place.

Responses: Stars in figure 1 were checked.

11-  In the discussion section mention how much decrease in ADG, DMI and feed digestion is caused by GINs/ Give some numbers.

Responses: Similar to above comment, we understand these numbers are very important for readers. But it hard to give a concision number, because the reported results were different due to different infection levels, animals species and feeding system. The number of ADG increase in the present study was added.

12-  In the discussion section mention how much increase in FCR is caused by GINs/ Give some numbers.

Responses: The numbers of changes in FCR and DMI in the present study was added in line 201-202.

13-  In the discussion section mention what is the mortality rate of GINs?

Responses: The number of mortality rate of sheep and goats was added according to reported Survey.

14- line 185 Revise figure 1. Some of stars were moved to the wrong place. 

Responses: Revised as suggested.

15- line189 How much decrease in ADG, DMI and feed digestion? Give some numbers. 

Responses: Revised as suggested.
16- line 190 increase FCR: How much decrease in ADG, DMI and feed digestion? Give some numbers.

Responses: Added as suggested.

17- line 190 what is the mortality rate?

Responses: There was no death in the present experiment.

We confirm that all comments were addressed to revise the manuscript. The manuscript is now for further review.

Thank you so much,

Regards,

Rongzhen Zhong (on behalf of myself and my all co-authors)

Reviewer 2 Report

The manuscript is of interest to the scientific community, and it is well written, although english should be revised by a native English speaker.

I recommend to accept it after revision of the english language.

Author Response

Dear reviewer,

Thank you for your review time which was put into this manuscript. We have invited an English speaking scientist to improve the language. 

We confirm that all comments were addressed to revise the manuscript. The manuscript is now for further review. 

Regards,

Rongzhen Zhong (on behalf of all authors)

Reviewer 3 Report

In this manuscript the authors were studying the effect of garlic supplementation on the parasite burden in lambs. The study was designed correctly and the proper controls were used. 

My remarks are: the authors could put more effort into data presentation. The bar diagrams look overcrowded: the error bars are off, the axis labels are too close the axes. 

table 3 is too crowded. The readability will improve if the authors convert it to the bar diagrams or two separate diagrams.

The statistical methods used throughout the paper have to be clearly described. What is the exact methods for a key findings of the paper :fecal egg count? 

Author Response

Animals

March 5, 2019

Dear reviewer:

Thanks for your detailed comments and review time which was put into this manuscript. Our responses to each comment are as follows:

In this manuscript the authors were studying the effect of garlic supplementation on the parasite burden in lambs. The study was designed correctly and the proper controls were used. 

1.     My remarks are: the authors could put more effort into data presentation. The bar diagrams look overcrowded: the error bars are off, the axis labels are too close the axes. 

Responses: The figure was improved. It seems that the changes were caused by typesetting.

2.     table 3 is too crowded. The readability will improve if the authors convert it to the bar diagrams or two separate diagrams.

Responses: Because the data was analyzed by mixed model of SAS and concluded two fixed effects (treatment and sampling day), so the table looks crowded. It will be even more crowded if the data were converted into bar diagrams or separate diagrams. So we prefer leave it in the present status.

3.     The statistical methods used throughout the paper have to be clearly described. What is the exact methods for a key findings of the paper :fecal egg count?

Responses: Please see Statistical analysis part for the statistical methods. We have analyzed many similar data in many previous papers (Zhong et al., Meat Science, 2015, 105,1-7.; Zhong et al., Animal Feed Science and Technology, 2018, 242, 127-134.). The method we used has been reviewed by a statistical expert (Shaoxun Tang, Region, Institute of Subtropical Agriculture, The Chinese Academy of Sciences, Changsha, Hunan, 410125, China, [email protected]).

  The main finding of this paper was that feeding garlic powder increased growth performance of lambs infected with GINs by decreasing fecal egg counts and improving feed digestion, rumen fermentation and health status of lambs. Because as for lambs infected with GINs, decrease in GIN infection is the direct way to improve performance and health status. So, the study determined the fecal egg counts.

We confirm that all comments were addressed to revise the manuscript. The manuscript is now for further review.

Regards,

Rongzhen Zhong (on behalf of myself and my all co-authors)
